# Changes in the Activity of the *CLE41/PXY/WOX* Signaling Pathway in the Birch Cambial Zone under Different Xylogenesis Patterns

**DOI:** 10.3390/plants11131727

**Published:** 2022-06-29

**Authors:** Natalia A. Galibina, Yulia L. Moshchenskaya, Tatiana V. Tarelkina, Olga V. Chirva, Kseniya M. Nikerova, Aleksandra A. Serkova, Ludmila I. Semenova, Diana S. Ivanova

**Affiliations:** Forest Research Institute, Karelian Research Centre of the Russian Academy of Sciences, 11 Pushkinskayast., 185910 Petrozavodsk, Russia; tselishcheva.yulia@mail.ru (Y.L.M.); karelina.t.v@gmail.com (T.V.T.); tchirva.olga@yandex.ru (O.V.C.); knikerova@yandex.ru (K.M.N.); serkovaaleksandra1996@yandex.ru (A.A.S.); mi7enova@gmail.com (L.I.S.); dszapevalova@mail.ru (D.S.I.)

**Keywords:** disturbance of xylogenesis, conducting phloem, cambial zone, differentiating xylem, *Cle41/44*, PXY, *WOX4*

## Abstract

The balance between cell proliferation and differentiation into other cell types is crucial for meristem indeterminacy, and both growth aspects are under genetic control. The peptide-receptor signaling module regulates the activity of the cambial stem cells and the differentiation of their derivatives, along with cytokinins and auxin. We identified the genes encoding the signaling module *CLE41-PXY* and the regulator of vascular cambium division *WOX4* and studied their expression during the period of cambial growth in the radial row: the conducting phloem/cambial zone and the differentiating xylem in two forms of *Betula pendula*, silver birch and Karelian birch. We have shown that the expression maximum of the *BpCLE41/44a* gene precedes the expression maximum of the *BpPXY* gene. Non-figured Karelian birch plants with straight-grained wood are characterized by a more intensive growth and the high expression of *CLE41/44-PXY-WOX4*. Figured Karelian birch plants, where the disturbed ratio and spatial orientation of structural elements characterizes the wood, have high levels of *BpWOX4* expression and a decrease in xylem growth as well as the formation of xylem with a lower vessel density. The mutual influences of *CLE41-PXY* signaling and auxin signaling on *WOX4* gene activity and the proliferation of cambium stem cells are discussed.

## 1. Introduction

Radial or secondary growth promotes the formation of vascular and mechanical stem tissue. The vascular tissues ensure the transport of water, minerals, photoassimilates, and signal molecules over long distances, while the mechanical tissues ensure the maintenance of the tree in an upright state, its resistance to wind, storms, etc. Radial growth is essential for the growth of woody plants and also plays a significant role in the long-term conservation of carbon in terrestrial biomass. The whole variety of structural elements of stem tissues comprises the lateral meristem—the cambium. Vascular cambium generates the initial vascular cells and their immediate derivatives, called mother cells [1]. Although the mother amplifying cells are anatomically indistinguishable from cambial stem cells, only stem cells retain the ability to generate both secondary phloem and secondary xylem elements. These mother amplifying cells have stably gained the cell fate of one of these tissues; they may differentiate directly into different tissues or may divide several times before terminal differentiation [1,2,3,4].

The balance between cell proliferation and differentiation into other cell types is crucial for meristem indeterminacy, and both growth aspects are under genetic control [5]. CLE (CLAVATA3 (CLV3)/EMBRYO SURROUNDING REGION-RELATED) peptides of group B (CLE41, CLE44 and CLE42) are among the main development regulators of lateral meristems, along with cytokinins and auxins [5,6,7,8,9,10,11,12,13,14,15]. The first CLE-peptide involved in the control of cambium activity was isolated from the *Zinnia elegans* culture as a factor stimulating the proliferation of cambial cells and inhibiting the differentiation of vascular elements, which was called TDIF (Tracheary Element Differentiation Inhibitory Factor) [7,8,9,16]. TDIF is encoded by the *CLE41* and *CLE44* genes in the *Arabidopsis* genome, which are expressed mainly in the phloem tissue and neighboring cells [7,8,9,16]. Upon cleavage and modification, TDIF is presumably released into the apoplastic space and diffuses toward the cambial cells, where it is bound by the plasma membrane-associated leucine-rich repeat receptor-like kinase (LRR-RLK) protein PHLOEM INTERCALATED WITH XYLEM (PXY)/TDIF RECEPTOR (TDR) [2,7,8]. Interactions between the peptide ligand TDIF/CLE41 and the TDR/PXY receptor have three independent effects on vascular development-related processes: (1) they promote cambial cell proliferation in the procambium/cambium; (2) inhibit xylem cell differentiation; and (3) control vascular patterning [2,5,6,8,9,17,18]. 

*WUSCHEL-RELATED HOMEOBOX4* (*WOX4*) and *WOX14* genes involved in the regulation of cambium cell proliferation are the identified targets of the TDIF/TDR signaling pathway [2,5,8,9,10,11,19,20,21]. The *CLE-PXY-WOX* signaling module is important for cambium growth and development, as shown in *Arabidopsis* plants. Cell division orientation depends on CLE41-peptide localization: the concentration gradient TDIF is essential at the beginning of vascular development, while *CLE41* expression should be limited to the phloem and adjacent cambium cells [7,13,22].

Data on the role of TDIF and their receptors (TDR) in regulating the secondary growth in woody plants are thus far scant. Such studies are not only of purely academic but also of applied value. The co-overexpression of *CLE41* and *PXY* genes in hybrid aspen under tissue-specific promoters corresponding to the original expression domains of both genes increases wood formation [13]. Other experiments have shown that *WOX4* plays a major role in controlling the cell identity and division activity in the vascular cambium of hybrid aspen [23]. The downregulation of *WOX4* homologs by RNA interference in hybrid aspen causes more dramatic phenotypic changes than are observed in annual species; in the most extreme cases, the resulting reductions in cambial activity and wood formation are severe enough to prevent the trees from remaining upright [2].

Most information on the roles of *CLE41/44* and *PXY* genes comes from studies involving their overexpression in plants or peptide treatment assays [2]. The objects of the present study were two forms of silver birch differing in wood figure: silver birch (*B. pendula* var. *pendula*), which forms a typical straight-grained wood, and a form of the silver birch, Karelian birch (*B. pendula* var. *carelica* (Merckl.) Hämet-Ahti) with figured wood. When the figured wood of Karelian birch is formed, the development program of vascular cambium cell changes: the program of cell death, leading to the formation of vessels and fibers of the xylem and sieve elements of the phloem, does not start, while the differentiating cambial derivatives preserve the protoplast and turn into storage parenchyma cells, which accumulate large amounts of storage substances [24,25,26]. The unique fact that different scenarios of xylogenesis can be studied within the same trunk makes Karelian birch a unique object for studying the mechanisms of wood formation [25,27,28]. 

In our previous work, we identified the genes encoding the main enzymes for auxin biosynthesis, transport, and conjugation and studied their expression models in various xylogenesis scenarios in the silver birch. We have shown that the auxin-deficient phenotype in Karelian birch trunk tissues is formed against the background of the overexpression of genes involved in auxin biosynthesis (*Yucca*), polar auxin transport (*PIN*), and the conjugation of auxin with amino acids (*GH3*) and UDP-glucose (*UGT84B1*) [26,29]. Since the TDIF/TDR-signaling influences the fate of the cambial initials and is closely related to auxin-signaling, in the present study, we focused on investigating the *CLE41/44-PXY-WOX4* gene expression in the cambial zone and differentiating xylem in two forms of *B. pendula:* silver birch and Karelian birch.

## 2. Results

The study objects were two forms of silver birch: *B. pendula* var. *pendula*, the form of silver birch with a straight-grained wood (hereafter Bp trees), and another silver birch variety, *B. pendula* var. *carelica*—Karelian birch. We selected Karelian birch trees possessing a highly figured wood (figured *B. pendula* var. *carelica* trees, future Bc FT trees) and non-figured plants that had a typical straight-grained wood with a weakly expressed figure (non-figured *B. pendula* var. *carelica* trees, hereafter Bc NF trees). Previously, we showed that the studied trees differed in the structure of mature xylem. The xylem in Bp trees displayed a typical for the species ratio of structural elements [30]. The xylem in Bc FT was characterized by low vessel density, a high proportion of parenchyma, and fibrous tracheids with thick cell walls. The straight-grained wood of the Bc NF trees was characterized by a high density of vessels [30].

### 2.1. Description of 1 and 2 Fractions in Three Phenotypes at Different Dates of Sampling

On 28 May, fraction 1 (from the bark side) in all three groups of trees included the cambial zone, early conducting phloem (part of the conducting phloem developed in the beginning of the growing season), and a small portion of non-conducting phloem (Figure 1a–c). On 11 June, fraction 1 of the figured Karelian birch species also contained three tissues: the cambial zone, the early conductive phloem, and a small portion of the non-conductive phloem (Figure 1f). In the silver birch and non-figured Karelian birch trees, on this date, fraction 1 contained several layers of differentiating xylem closest to the cambium, aside from the three tissues above-mentioned (Figure 1d,e). The reason for the different composition of tissues in fraction 1 was the more active division of the cambium toward the xylem in two groups of trees that formed normal wood. On 26 June, fraction 1 included five tissues: several layers of the differentiating xylem closest to the cambium, the cambial zone, early conducting phloem, differentiating late conducting phloem (part of the conducting phloem developed in the second half of the growing season), and a small portion of non-conducting phloem in all three groups of trees (Figure 1g–i). 

Fraction 2 (from the side of the wood) included a differentiating xylem at all periods of sampling and contained xylem cells at the stage of expansion and the formation of a secondary cell wall (Figure 1).

### 2.2. The Anatomical Features of the Trunk Tissues in Different Forms of Silver Birch

A few differentiating xylem cells were observed (Figure 2a) in the Bp trees (dated May 28). Every fortnight, the width of the current year xylem increment increased by 400 and 500 µm by 11 June and 25 June, respectively (Figure 2b,c). If on 11 June the entire xylem of the current year was in a differentiating state (Figure 1d), then on 25 June, most of the xylem of the current year was mature xylem (Figure 1g).

The Bc NF trees (dated May 28) were characterized by the most active divisions of the cambium toward the xylem compared to the Bp trees (Figure 1a,b and Figure 2a). Over the next two weeks, the xylem increment was maximal, and xylem width increased by 580 µm by 11 June and by 450 µm by 25 June (Figure 2b,c). 

The Bc FT trees were characterized by the absence of the current year xylem increment (28 May) and less xylem width (11 June) compared to the Bp and Bc NF trees (Figure 1c,f and Figure 2a,b). The increment of xylem in the current year did not significantly differ from those in the Bp and Bc NF trees by 25 June (Figure 2c).

We failed to count the number of cells in the cambial zone of the samples collected on May 28 because thin-walled and highly hydrated cells of the cambium ruptured during sampling. On 11 June, the number of cells in the cambial zone did not differ in all three studied phenotypes. Two weeks later (25 June), the number of cells of the cambial zone in the Bp and Bc NF trees decreased, especially in the Bc NF trees, while the Bc FT trees remained at the same level (Table 1).

As we had already observed a significant increase in the conductive phloem width in the Bp trees on 28 May, by 11 June, its value did not change and increased by 26 μm by the end of June (Figure 3). The number of differentiating phloem cells at the end of May and the middle of June was the same and increased only at the end of June. The number of sieve tubes increased by 11 June and remained at this constant level until the end of June (Table 1). 

From 28 May to 11 June in the Bc NF trees, the width of the conducting phloem increased by 30 μm, and from 11 June to 25 June by 12 μm. The conducting phloem was wider in the Bc NF trees compared to the Bp trees only on 11 June (Figure 3). The number of differentiating phloem cells increased from 28 May to 25 June. Their number in the Bc NF trees was less than that of the Bp trees in May, and did not differ from the Bp trees on 11 June. The number of sieve tubes increased by 11 June and remained constant until 25 June, like in the Bp trees (Table 1). 

The features of the Bc FT trees were the smaller width of the conducting phloem (Figure 3) and the smaller number of sieve tubes (Table 1) compared to the Bp and Bc NF trees. The number of differentiating phloem cells on the different dates of sampling was similar to the Bc NF trees.

### 2.3. CLE41/44 Gene Identification in the Silver Birch Genome

Genome-wide analysis followed by manual validation and the removal of false positives allowed us to identify two *B. pendula* genes encoding putative CLE41/44 proteins (Table 2). One gene was on chromosome 1, and the second on contig 443. Both genes contained one exon and had no introns (Figure 4).

We found that the number of *BpCLE41/44b* transcripts was close to zero or not detected in all samples studied. Similar data were obtained by J. Alonso-Serra et al. [31] when studying the molecular fingerprints of eight main tissue types in the stem of *B. pendula*: phellem, combined phellogen and phelloderm, nonconductive secondary phloem, conductive phloem, cambium, developing xylem, xylem, and last year’s xylem tissue. The analysis of the normalized transcript abundances (TPM) for each tissue showed scanty amounts of *BpCLE41/44b* (*Bpev01.c0443.g0012.m0001*, Table S2 from [31]) in the conductive phloem and cambium. Only the *BpCLE41/44a* data will be discussed in this paper.

### 2.4. PXY Gene Identification in the Silver Birch Genome

Seven genes encoding putative receptor such as kinases homologous to TDR/PXY of other plant species (*Arabidopsis thaliana* and *Populus trichocarpa*) were identified in the *B. pendula* genome (Table 3). The genes were on four different chromosomes and three contigs. All genes contained two exons, except for *Bpev01.c1696.g0001*, which contained three exons (Figure 5).

The phylogenetic analysis revealed that gene *Bpev01.c0668.g0016* encoded the protein, which was highly similar to PXY (Figure 6). The sequence identity (in percentage of the total number of residues) for the *A. thaliana* PXY and *B. pendula* PXY protein pair was 60.5%.

### 2.5. WOX Gene Identification in the Silver Birch Genome

Eleven genes encoding putative homologous to WUSCHEL-related homeobox (WOX) transcription factors of other plant species (*Arabidopsis thaliana* and *Populus trichocarpa*) were identified in the *B. pendula* genome (Table 4). The genes were on seven different chromosomes and contained two to four exons (Figure 7).

Phylogenetic analysis performed on protein sequences revealed that the gene *Bpev01.c0640.g0004* encoded the protein, which was highly similar to *WOX4* (Figure 8). The sequence identity (in percentage of the total number of residues) for the *A. thaliana WOX4* and *B. pendula*
*WOX4* protein pair was 55.6%.

### 2.6. Expression of the Genes Encoding CLE-PXY-WOX Signaling Pathway in the Trunk Tissues of Bp Trees during Cambial Growth

We have previously shown that *WOX4* is a cambial stem cell pattern, and its activity correlates with the proliferation process [23,32]. Therefore, we used the expression level of *BpWOX4* in the studied fractions to estimate cell proliferation. We found that the number of *BpWOX4* transcripts in the Bp trees did not differ significantly between Fractions 1 and 2. The expression level of *BpWOX4* was higher in the differentiating xylem on 28 May and 25 June, while on 11 June, it was higher in Fraction 1 in contrast (Figure 9a). We assumed that cell proliferation was observed in Fractions 1 and 2 on all sampling dates.

The WOX4 homeodomain transcription factors are downstream targets of the TDIF/CLE41-PXY/TDR signaling pathway [9,32]. We showed that the maximum expression of *BpCLE41/44a* in the Bp trees was in tissues sampled from the bark side on 28 May, and its value was 10 times higher than that from the xylem side (Figure 9d). Previously, J. Alonso-Serra et al. [31] found the TPM maximum for *BpCLE41/44a* in non-conductive secondary phloem, conductive phloem, and cambium (*Bpev01.c0016.g0065*, Table S2 after [31]). At the middle and end of June, the number of *BpCLE41/44a* transcripts did not differ between Fractions 1 and 2. On 11 June, the expression level of *BpCLE41/44a* was two times higher than on June 25 (Figure 9d).

We showed that the number of *BpPXY* transcripts from the side of differentiating xylem compared to the tissue layer including the cambial zone and vascular phloem was higher in all trees studied. We observed the maximum at the middle and end of June (Figure 9g).

### 2.7. The Formation of Straight-Grained Wood of Bc NF Trees Occurs against the Background of Greater Expression of the Genes CLE-PXY-WOX in the Cambial Zone and Differentiating Xylem Compared to the Bp Trees

The maximum expression level of the genes *CLE-PXY-WOX* in the Bc NF trees was at the same tissues and time as in the Bp trees: for *BpCLE41/44a*, it was the date of 28 May, Fraction 1 (Figure 9e); for *BpPXY*, it was 11 June, Fraction 2 (Figure 9h); and for *BpWOX4*, 28 May, Fraction 2 (Figure 9b). We found that in the Bc NF trees, the expression level of *BpCLE41/44a* on 28 May on the bark side was five times greater compared to the Bp trees (Figure 9e), and the expression level of *BpPXY* (11 June) on the differentiating xylem side was three times greater (Figure 9h). The proliferation of cells (28 May) in Fraction 2 in the Bc NF trees was approximately two times higher than that in the Bp trees. The dates of 11 June and 25 June were characterized by the fact that the number of *BpWOX4* transcripts decreased and did not differ between the two fractions (Figure 9b).

### 2.8. Decreased Expression of the Genes CLE-PXY When Forming the Wood Figure in Bc FT Trees

In the Bc FT trees, the number of *BpCLE41/44a* and *BpPXY* transcripts was lower compared to the other forms. Similar to the Bp and Bc NF trees, the maximum expression of *BpCLE41/44a* was observed in Fraction 1 (28 May) (Figure 9f); for *BpPXY*, it was the date of 11 June in Fraction 2 (Figure 9i). The lower expression of the genes *CLE-PXY* in the Bc FT trees, compared to the Bp and Bc NF trees, may be responsible for (1) the absence of a current year xylem increase (28 May) and the lower xylem increase (11 June) (Figure 1c,f and Figure 2a,b); and (2) the smaller width of the conducting phloem (Figure 3) and the smaller number of sieve tubes (Table 1). 

Another specific feature of the Bc FT trees is the maximum cell proliferation from the xylem side (11 June), while the expression level of *BpWOX4* significantly exceeded those of the Bp and Bc NF trees (Figure 9c).

## 3. Discussion

Plant biomass is a huge renewable resource of biofuels and biomaterials, and the major constituent of plant biomass is xylem derived from the vascular meristem [33]. Therefore, much attention has recently focused on the study of the molecular mechanisms of xylogenesis. In studies performed on *Arabidopsis* as the model plant, it has been shown many times that the maximum expression of *CLE41/44* is observed in the phloem tissue and neighboring cells [7,9,16], while *PXY* and *WOX4* are in part of the xylem-facing cambium [3,4]. We have shown that during the period of cambial growth in silver birch, the expression of the gene *CLE-PXY-WOX* patterns were extended in a radial row «the conductive phloem/the cambial zone (Fraction 1)—the differentiating xylem (Fraction 2)» with the maximum for each gene in that or another faction. The expression of the genes changed during cambial growth (28 May–25 June).

We found that the highest expression of the gene *BpCLE41/44a* was observed on 28 May, then decreased in all of the studied birch phenotypes (Bp, Bc NF, and Bc FT trees). In contrast, the expression levels of the gene *BpPXY* were the highest on 11 June. Previously, it has been shown many times in the literature that the spatial separation of the ligand (TDIF) and receptor (TDR) is necessary for the correct spatial orientation of the vascular pattern and the restriction of the stem cell zone [33]. We have shown that, together with spatial separation (Fraction 1 for *BpCLE41/44a* and Fraction 2 for *BpPXY*), we also observed temporal separation (*BpCLE41/44a* and *BpPXY* maximum at different stages of cambial growth).

In the Bp and Bc NF trees, forming straight-grained wood, the maximum expression of the gene *BpWOX4* matched in time to *BpCLE41/44a*. Unlike *BpCLE41/44a*, whose peak activity was higher in Fraction 1, the maximum number of *BpWOX4* transcripts was observed in Fraction 2. We have shown that the Bc NF trees, compared to Bp trees, are characterized by (1) greater expression of the genes *BpCLE41/44a*, *BpPXY*, *BpWOX4*; and (2) the most active divisions of the cambium toward the xylem. In our previous studies, we characterized the phenotype of the Bc NF trees as faster growing than the Bp trees. The straight-grained wood of the Bc NF trees was characterized by a high density of vessels [30].

We found that in the abnormal areas of the Karelian birch trunk, the maximum expression of the gene *BpWOX4* did not coincide in time with the maximum of *BpCLE41/44a* and was observed on 11 June. The number of *BpWOX4* transcripts significantly outnumbered those in the Bp and Bc NF trees. The expression level of the genes *CLE41/44-PXY* running *WOX4* expression [9,32] in the Bc FT trees was significantly lower compared to the Bp and Bc NF trees.

We know that *WOX4* is considered as a central regulator of vascular cambium division because it activates a cambium-specific transcriptional network and integrates TDIF/TDR signaling and auxin signaling for cambium division [9,20,34,35,36]. If TDIF/TDR signaling stimulates the *WOX4* transcription and promotes the cambium proliferation in stems [9], then auxin signaling attenuates the activity of the stem cell-promoting *WOX4* gene, and cell-autonomously restricts the number of stem cells in stems [34].

The TDIF/TDR signaling, aside from stimulating *WOX4* transcription and promoting the cambium proliferation in stems [9], leads to the inactivation of the BES1 [37,38] and BIL1 [37,39] transcription factors. The BES1 (BRI1 (BRASSINOSTEROID INSENSITIVE)-EMSSUPPRESSOR) 1) transcription factor promotes xylem identity and also suppresses the expression of *WOX4*. The interaction between TDR and GSK3 (Glycogen Synthase Kinase 3) proteins (BIN2) is thought to result in the negative regulation of the BES1 by phosphorylation [37]. The BIL1 (BIN2 (BRASSINOSTEROID INSENSITIVE 2)-LIKE 1) refers to GSK3-proteins. The BIL1 links the TDIF/TDR module with auxin–cytokinin signaling during secondary growth. The BIL1 functions as a negative regulator of cambial activity downstream of the TDR by targeting the ARF5 (auxin response factor) protein [37]. Phosphorylation of ARF5 by BIL1 weakens its binding to the repressor Aux/IAA (Aux/INDOLE-3-ACETIC ACID) and enhances the negative effect ARF5 on the activity of vascular cambial, which upregulates *ARR7* (ARABIDOPSIS RESPONSE REGULATOR 7) and *ARR15* (two negative regulators of cytokine signaling). The BIL1 activity is inhibited by TDR, attenuating the effect of ARF5 on the expressions of *ARR7* and *ARR15* and increasing the vascular cambial activities [36,39,40]. High concentrations of auxin promote ubiquitin-mediated proteolysis of Aux/IAA transcriptional repressors, releasing ARF5 to regulate xylem differentiation. ARF5 autonomously limits the number of stem cells through the weakening of *WOX4* activity and increases the expression of the *HB8* gene, positively affecting xylem differentiation [2,41] (Figure 10).

From this, it follows that (1) the presence of a high expression of *CLE41/44-PXY* and/or (2) reduced auxin signaling is necessary to maintain a high expression of the gene *BpWOX4*. We have previously investigated the expression level of *BpARF5* and its direct target, *BpHB8*, in the trunk tissues of different forms of silver birch [30]. We found that the level of *BpARF5* expression in the differentiating xylem of Karelian birch plants with both figured and non-figured wood was lower than in the Bp trees. The formation of the figured wood of the Bc FT trees was associated with the low expression of *BpHB8*, whereas in the Bc NF trees with a straight-grained wood, the expression of *BpHB8* was indistinguishable from that of the Bp trees [30]. Collectively, the data obtained through this study and previous ones allow us to propose that the maximum expression of the gene *BpWOX4* in Bc FT trees on 11 June was due to (1) continuing TDIF/TDR signaling and (2) reduced auxin signaling. This suggestion is consistent with the anatomical evidence of lower xylem increases in the Bc FT trees compared to the Bp and Bc NF trees.

## 4. Materials and Methods

### 4.1. Study Objects

The study objects were two forms of silver birch: *Betula pendula* Roth var. *pendula*—the form of silver birch with straight-grained wood, and *B. pendula* var. *carelica*—Karelian birch. All plants grew in the same soil and climatic conditions on the experimental plot of the KarRC RAS near Petrozavodsk city. The trees were 14–15 years old. The Karelian birch plants were grown from seeds obtained from the controlled pollination of Karelian birch plus trees. The number of trees in each group was nine. The samples were collected during the period of cambial growth (28 May 2020, 11 June 2020, and 26 June 2020).

### 4.2. Plant Sampling

The trunk tissue samples were taken at breast height (1.3 m above ground level). The sections of the trunk with the highest degree of manifestation of structural anomalies were selected for the sampling of plant material from the Karelian birch trunks. For microscopic analysis blocks including the phloem, the cambial zone and the last 2–3 annual increments of wood were cut out (5 × 5 × 3 mm, length x width x height). For molecular genetic analysis, «windows» of 6 × 8 cm were cut out of the trunk and the bark was separated from the wood (Figure 11). During the period of cambial growth, the bark moves away from the wood along the expanding xylem zone. Tissue complexes (hereafter Fraction 1) were prepared from the inner surface of the bark. The layers of tissue (hereafter Fraction 2) were scraped off the exposed wood surface with a blade. The sampling of stem tissues was monitored under a light microscope (Figure 1). The material was frozen in liquid nitrogen and stored at −80 °C. 

### 4.3. Microscopy

Three samples of phloem and xylem tissues from each tree were analyzed. Sample preparation for microscopy was conducted as described previously [26]. The samples were fixed in glutaraldehyde, dehydrated in a series of alcohols of rising concentrations, and embedded in an Araldite-Embed-812 mixture following the published technique [42]. Cross sections 2 µm thick were cut with an Ultrotome IV (LKB, Bromma, Sweden) and stained with a 1% aqueous solution of safranin. Permanent slides were made using synthetic mounting medium Vitrogel (BioVitrum, St. Petersburg, Russia). Microscopic analysis was carried out under an AxioImager A1 light microscope (Carl Zeiss, Jena, Germany) equipped with an ADF PRO03 camera. Images were processed with ADF Image Capture software (ADF Optics Ningbo, China). Anatomical measurements were made following the available guidelines using panoramic cross sections with an area of 7–10 mm^2^ [43,44,45].

### 4.4. Gene Retrieval from the Silver Birch Genome by Bioinformatics Methods

The search for the *CLE41/44*, *PXY*, and *WOX4* genes was carried out using the published genome of *Betula pendula* Roth [46]. To this end, the CDS of *Arabidopsis thaliana* and *Populus trichocarpa CLE41/44*, the *PXY* and *WOX4* genes, and the amino acid sequences of the corresponding proteins were obtained from The Arabidopsis Information Resource (TAIR) database (release 13, https://www.arabidopsis.org, accessed on 20 April 2020) and the Phytozome database (http://www.phytozome.net/poplar, release v3.0, accessed on 20 April 2020), respectively. The resulting sequences were then used as a BLAST search query across the genome of *B. pendula* var. *pendula* (release 1.2, https://genomevolution.org/coge, accessed on 20 April 2020) to identify homologous sequences.

The structures of candidate proteins were predicted using the National Center for Biotechnology Information (NCBI) resource (http://www.ncbi.nlm.nih.gov/Structure/cdd/cdd.shtml, accessed on 20 April 2020) [47]. The prediction of protein subcellular localization was performed using DeepLoc 2.0 [48]. Phylogenetic analysis was carried out using MEGA X software [49]. Multiple sequence alignment of the protein sequences was performed using ClustalW. Phylogenetic trees were constructed using the neighbor-joining method based on the Poisson correction model with 1000 bootstrap replicates [50,51,52]. The different genes in *B. pendula* were named according to the Phylogenetic analysis. The percent identity of *B. pendula* and *A. thaliana* proteins was determined using the EMBOSS Needle online tool (https://www.ebi.ac.uk/Tools/psa/emboss_needle/, accessed on 20 April 2020). The gene structures were analyzed using the Gene Structure Display Server (//gsds.gao-lab.org/, accessed on 20 April 2020) [53].

### 4.5. qRT-PCR

The isolation of total RNA was performed using an extraction CTAB buffer (pH 4.8–5.0): 100 mM Tris–HCl (pH 8.0), 25 mM EDTA, 2M NaCl, 2% CTAB, 2% PVP, and 2% mercaptethanol was added to the mixture before use. Plant tissue lysates were additionally treated with DNase and RNase inhibitors (Syntol, Moscow, Russia). Separation of the aqueous and organic phases was conducted using a mixture of chloroform–isoamyl alcohol (24:1). RNA was precipitated using 25 mM LiCl, then re-precipitation was carried out using an extraction SDS buffer: 1 M NaCl, 0.5% SDS, 10 mM Tris–HCl (pH 8.0), 1 mM EDTA [54]. RNA was re-precipitated with absolute isopropanol. The quality and quantity of the isolated RNA and synthesized cDNA were checked spectrophotometrically and by electrophoresis in 1% agarose gel.

Specific primers (Sintol, Moscow, Russia) for amplification of the studied genes were designed using the software Beacon Designer 8.21 (PREMIER Biosoft, San Francisco, CA, USA) (Table 5). As a reference gene for the normalization of quantitative PCR data, we used genes of the *Ef1a* and *GAPDH* family [55].

In previous studies, we evaluated the suitability of five genes—*GAPDH1*, *Actin1*, *Ef1a(1)*, *Ef1a(2)*, and *18SrRNA*—in two forms of silver birch for use as a reference when staging qRT-PCR based on the stability of their expression. The results of studying the level of expression of the potential reference genes were analyzed using the NormFinder and BestKeeper programs. We showed that among the studied genes, the most stably expressed in the *B. pendula* samples (leaves and trunk tissues) were *Actin1*, *GAPDH1*, and *Ef1a(1)*, while the *GAPDH1* and *Ef1a(1)* genes were suggested by NormFinder as the best combination of two reference genes and had the lowest stability index [55]. Families of genes coding for (1) peptide ligand TDIF/CLE41 (*BpCLE41/44a* and *BpCLE41/44b*); (2) the TDR/PXY receptor (*BpPXY*); and (3) WUSCHEL-RELATED HOMEOBOX4 (*BpWOX4*) involved in the regulation of cambium cell proliferation as the identified targets of the TDIF/TDR signaling pathway were studied. 

Amplification of the samples was performed using an iCycler instrument with an iQ5 optical module (Bio-Rad, Hercules CA, USA) and an amplification kit with an intercalating dye SYBR Green (Evrogen, Moscow, Russia). The specificity of the amplification products was checked by melting the PCR fragments. The relative quantity of gene transcripts (RQ) was calculated from the formula: RQ = E^−ΔCt^,(1)
where ΔCt is the difference in the threshold cycle values for the reference and target genes, and E is the effectiveness of PCR. To determine the efficiency (E), PCR amplification was performed with each pair of primers in a series of 10-fold dilutions (10^−1^, 10^−2^, 10^−3^, 10^−4^, and 10^−5^) of cDNA. Using Excel software, a plot of Ct versus Lg (conc. cDNA) was plotted, and using the values of the slope of the curve (slope, k), the efficiency was calculated using the formula [56]:E = (10^−1/K^ − 1) × 100.(2)

The level of expression of specific genes was expressed in relative units (arbitrary units).

### 4.6. Statistical Data Processing

The results were statistically processed with PAST (version 4.0). Before starting the statistical analysis, raw data were initially tested for normality using the Shapiro–Wilk test. The significance of differences between variants was estimated by the Mann–Whitney U-test. Different letters indicate a significant difference at *p* < 0.05. All data in the diagrams appear as mean ± SE, where SE is the standard error. Sample sizes are denoted as *n*.

The research was carried out using the equipment of the Core Facility of the Karelian Research Center of the Russian Academy of Sciences.

## 5. Conclusions

The distribution of the expression of the genes *CLE-PXY-WOX* was studied during the period of cambial growth in the radial row: the conducting phloem/cambial zone and the differentiating xylem under different scenarios of xylogenesis using two forms of silver birch as an example. We showed that the expression maximum of the *CLE41/44* and *PXY* genes are separated not only in space (conductive phloem/cambial zone (Fraction 1)—differentiating xylem (Fraction 2)), but also in time. The expression maximum of the *BpCLE41/44a* gene precedes the expression maximum of the *BpPXY* gene. TDIF/TDR signaling induces *BpWOX4* gene expression and stem cell proliferation in the cambium. Thus, the phenotype of the Bc NF trees with high expression of *CLE41/44-PXY-WOX4* is characterized by more intensive growth. Auxin signaling against the background of a decrease in the expression of the *CLE41/44* genes leads to a decrease in the expression of the *BpWOX4* genes, and, accordingly, the proliferation of cambial initials. The auxin-deficient phenotypes of Bc FT trees showed high levels of *BpWOX4* expression and a decrease in xylem growth as well as the formation of xylem with a lower vessel density. A significant decrease in free (physiologically active) auxin because of its conjugation with UDP-glucose and amino acids described previously [26,29] may lead to more dramatic scenarios in the Bc FT trees in the zones of development of structural abnormalities; the differentiating cambial derivatives preserve the protoplast and turn into the storage parenchyma cells.

## Figures and Tables

**Figure 1 plants-11-01727-f001:**
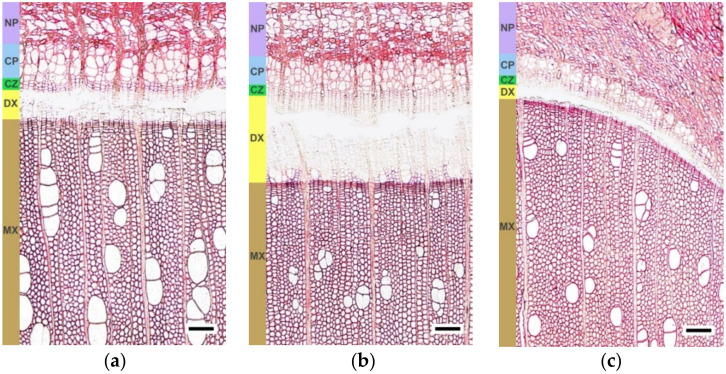
The transverse sections showing the stem tissues of *B. pendula* var. *pendula* (**a**,**d**,**g**), non-figured (**b**,**e**,**h**), and figured (**c**,**f**,**i**) *B. pendula* var. *carelica* trees. Samples were collected on 28 May 2020 (**a**–**c**), 11 June 2020 (**d**–**f**), 25 June 2020 (**g**–**i**). NP—non-conducting phloem; CP—conducting phloem; CZ—cambial zone; DX—differentiating xylem; MX—mature xylem. Scale bar = 200 µm.

**Figure 2 plants-11-01727-f002:**
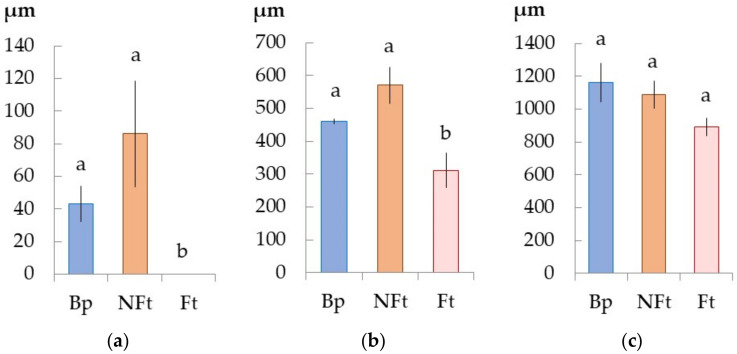
The current year xylem growth ring width of *B. pendula* var. *pendula* (Bp), non-figured (NFt), and figured (Ft) *B. pendula* var. *carelica* trees. Samples were collected on 28 May 2020 (**a**), 11 June 2020 (**b**), and 25 June 2020 (**c**). *n* = 9.

**Figure 3 plants-11-01727-f003:**
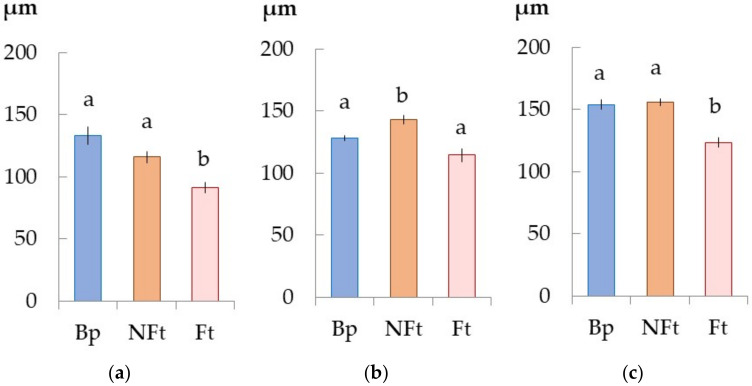
The width of the conductive phloem of *B. pendula* var. *pendula* (Bp), non-figured (NFt), and figured (Ft) *B. pendula* var. *carelica* trees. Samples were collected on 28 May 2020 (**a**), 11 June 2020 (**b**), and 25 June 2020 (**c**). *n* = 9.

**Figure 4 plants-11-01727-f004:**
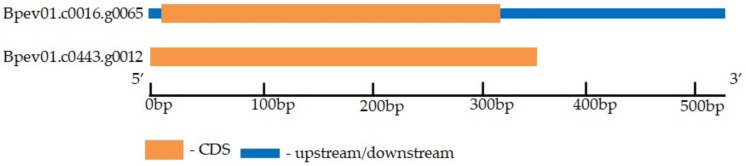
The structure of the *B. pendula*
*CLE41/44* genes. The exon and untranslated region (UTR) are represented by orange boxes and blue boxes, respectively.

**Figure 5 plants-11-01727-f005:**
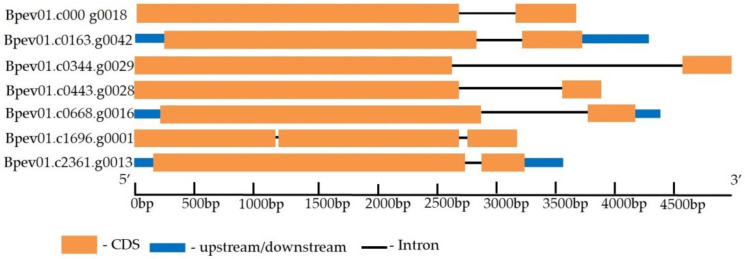
The structure of *B. pendula* genes encoding LRR-RLKs, homologous to PXY. The intron, exon, and untranslated region (UTR) are represented by black lines, orange boxes, and blue boxes, respectively.

**Figure 6 plants-11-01727-f006:**
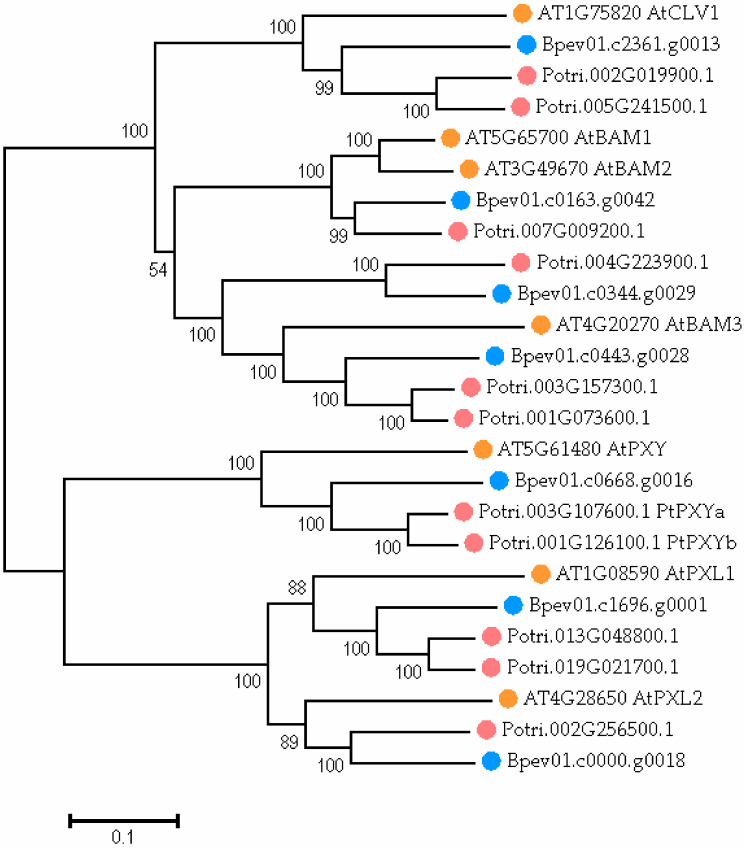
The phylogenetic tree of proteins from *Arabidopsis thaliana* (orange dots), *Populus trichocarpa* (pink dots), and *Betula pendula* (blue dots). The tree is drawn to scale, with branch lengths in the same units as those of the evolutionary distances used to infer the phylogenetic tree. The numbers shown next to the branches represent the results of the bootstrap test (1000 replicates). The access codes of the *A. thaliana* and *P. trichocarpa* proteins in the TAIR and Phytozome databases are indicated next to the corresponding proteins.

**Figure 7 plants-11-01727-f007:**
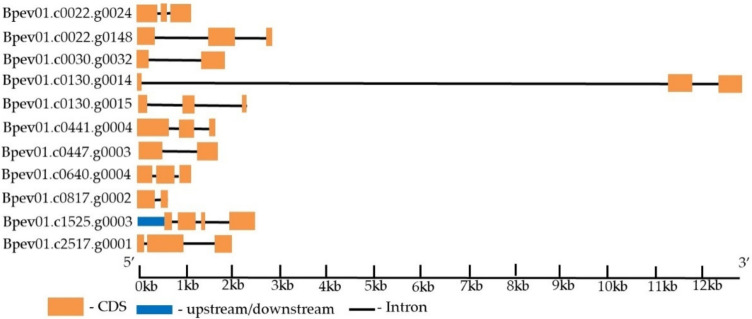
The structure of the *B. pendula*
*WOX* genes. The intron, exon, and untranslated region (UTR) are represented by black lines, orange boxes, and blue boxes, respectively.

**Figure 8 plants-11-01727-f008:**
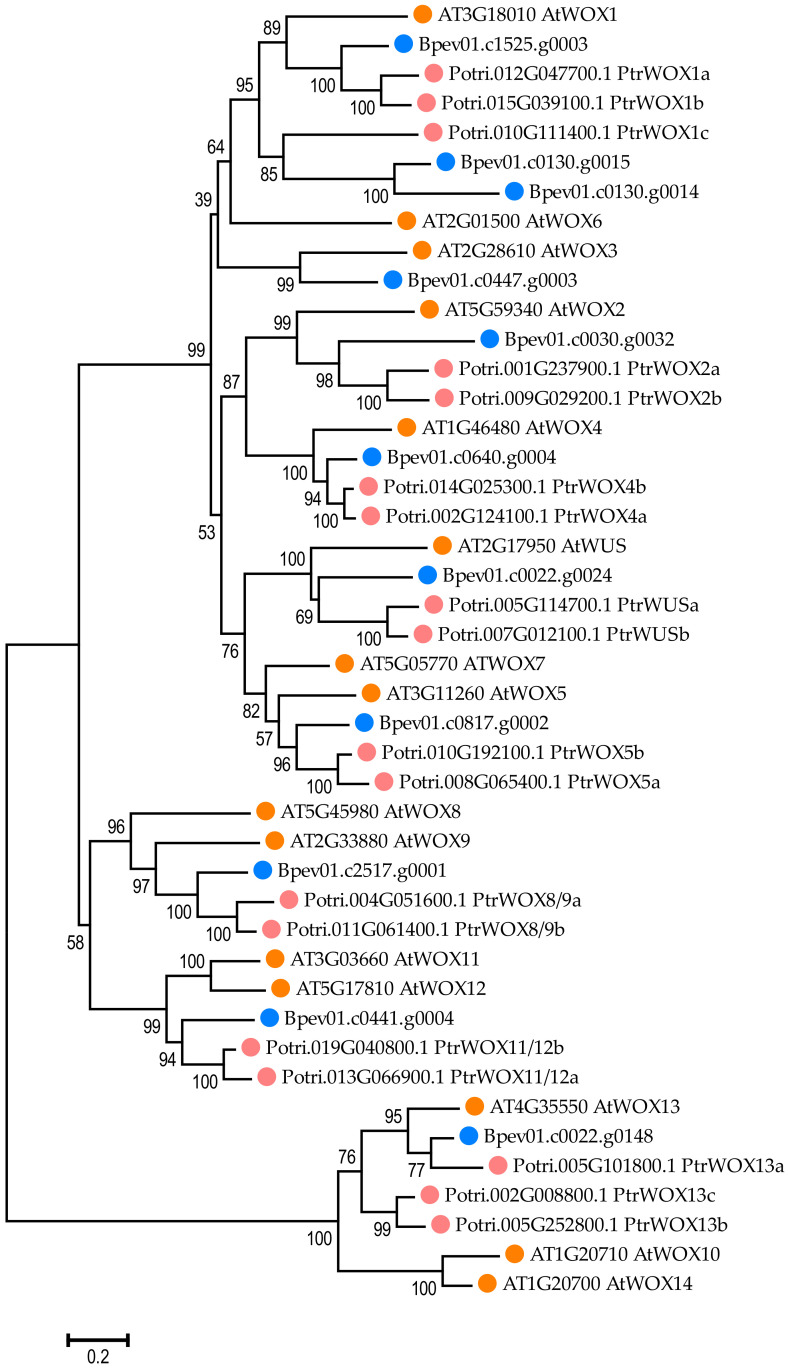
The phylogenetic relationships of the WOXs of *B. pendula* (dark blue dots), *A. thaliana* (orange dots), and *P. trichocarpa* (pink dots). The tree was drawn to scale, with branch lengths in the same units as those of the evolutionary distances used to infer the phylogenetic tree. The numbers shown next to the branches represent the results of the bootstrap test (1000 replicates). The access codes of the *A. thaliana* and *P. trichocarpa* proteins in the TAIR and Phytozome databases are indicated next to the corresponding proteins.

**Figure 9 plants-11-01727-f009:**
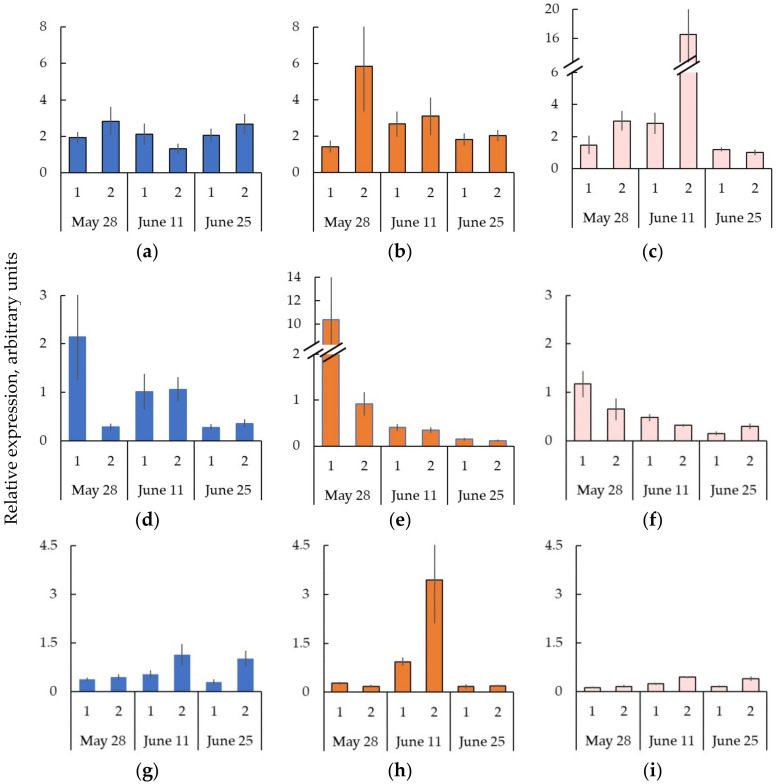
The relative expression (arbitrary units) of genes *BpWOX4* (**a**–**c**), *BpCLE41/44a* (**d**–**f**), and *BpPXY* (**g**–**i**) in tissues including the cambial zone, differentiating phloem, and mature phloem (1) and differentiating xylem (2) *B. pendula* var. *pendula* (**a**,**d**,**g**), non-figured (**b**,**e**,**h**), and figured (**c**,**f**,**i**) *B. pendula* var. *carelica* trees. Samples were collected on 28 May, 11 June, and 26 June 2020. *n* = 9.

**Figure 10 plants-11-01727-f010:**
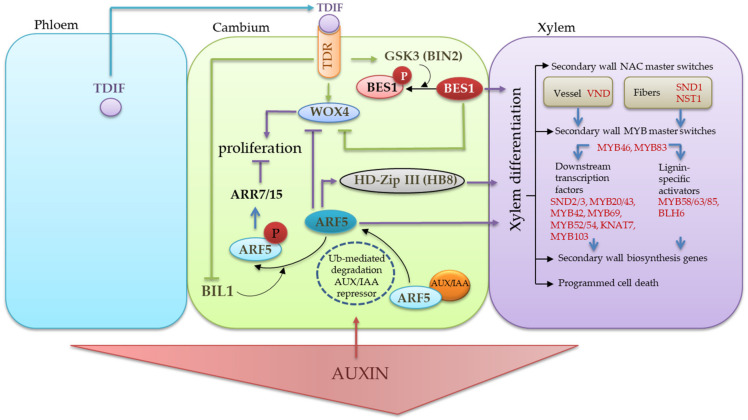
The schematic representation of the hormonal and molecular genetic regulation of cambium activity in woody plants. The TDIF signal peptide is produced in the phloem and migrates to the cambium, where it binds to its TDR receptor. The TDIF/TDR signaling activates *WOX4* by regulating cambium cell proliferation [4,12,23]. TDIF binding to TDR activates GSK3 (BIN2) and leads to the phosphorylation and degradation of the BES1 protein. BES1 promotes xylem differentiation and also suppresses *WOX4* expression, so its degradation preserves cambium pluripotency [37]. The TRD acts as a suppressor of another GSK3—BIN2-LIKE 1 (BIL1). BIL1 in the absence of TDR phosphorylates the auxin-dependent transcription factor of ARF5, releasing it by weakening the interaction of ARF5 with the AUX/IAA repressor [40]. Phosphorylation of ARF5 enhances its activity in suppressing the cytokine response through the expression of *ARR7/15* and leads to a decrease in cambium proliferation [39]. Auxin signal transduction in cambial stem cells is required to maintain cambial activity [20]. The level of auxin signal transmission increases in the differentiating xylem. High concentrations of auxin promote the ubiquitin-mediated proteolysis of Aux/IAA transcriptional repressors, releasing ARF5 to regulate xylem differentiation. ARF5 autonomously limits the number of stem cells by downregulating the *WOX4* activity and increasing the *HB8* gene expression, positively influencing xylem differentiation [2].

**Figure 11 plants-11-01727-f011:**
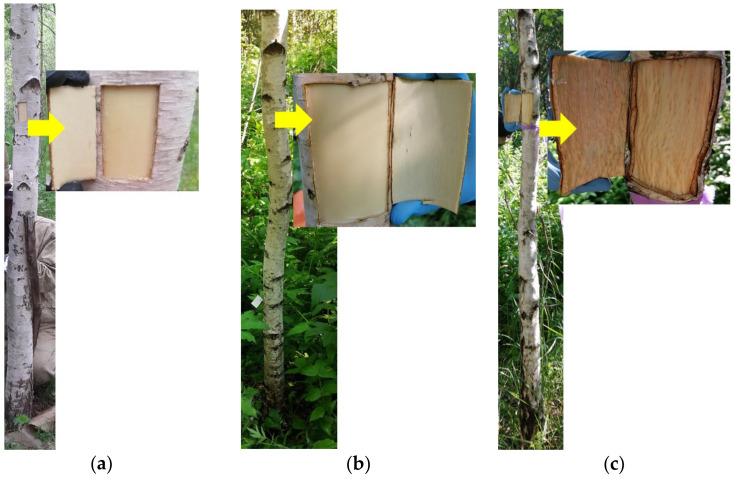
The trunk, debarked wood surface, and inner bark surface of *B. pendula* var. *pendula* (**a**), non-figured (**b**), and figured (**c**) *B. pendula* var. *carelica* trees.

**Table 1 plants-11-01727-t001:** The number of cells in the radial row: the cambial zone, the conducting phloem of *B. pendula* var. *pendula* (Bp), non-figured (NFt), and figured (Ft) *B. pendula* var. *carelica* trees. Samples were collected on 28 May 2020, 11 June 2020, and 25 June 2020.

Phenotype	Number of Cells of Cambial Zone		Number of Differentiating Phloem Cells			Number of Sieve Tubes		
	11 June	25 June	28 May	11 June	25 June	28 May	11 June	25 June
Bp trees	8.6 ± 0.3	7.9 ± 0.5	2.9 ± 0.1	2.9 ± 0.1	3.5 ± 0.1	3.8 ± 0.1	4.9 ± 0.1	4.9 ± 0.1
BcNFt trees	8.8 ± 0.3	6.3 ± 0.3	2.3 ± 0.2	3.0 ± 0.1	3.4 ± 0.1	3.6 ± 0.2	5.2 ± 0.2	5.2 ± 0.2
Bc FT trees	8.3 ± 0.3	8.3 ± 0.4	2.2 ± 0.2	3.3 ± 0.1	3.6 ± 0.2	3.2 ± 0.3	4.3 ± 0.2	3.1 ± 0.2

**Table 2 plants-11-01727-t002:** The features of the *B. pendula*
*CLE41/44* members.

Gene Name	Gene ID	Number of Amino Acid Residues in Prepropeptide	CLE-Motif Sequence	Subcellular Localization (Likelihood)
*BpCLE41/44a*	*Bpev01.c0016.g0065*	104	HEVPSGPNPISN	Extracellular (0.8902)
*BpCLE41/44b*	*Bpev01.c0443.g0012*	117	HEVPSGPNPISN	Extracellular (0.8565)

**Table 3 plants-11-01727-t003:** The features of the *B. pendula* subgroup XI receptor kinase-like (RLK) family members.

Gene Name	Gene ID	Number of Amino Acid Residues in Peptide	Location of Transmembrane Helices	Location of Protein Kinase Domain	Subcellular Localization (Likelihood)
*BpPXL2*	*Bpev01.c0000.g0018*	1020	642–664	714–985	Cell membrane (0.9555)
*BpBAM1/2*	*Bpev01.c0163.g0042*	1022	638–657	696–1006	Cell membrane (0.9967)
*BpBAM3a*	*Bpev01.c0344.g0029*	997	628–650	688–993	Cell membrane (0.9942)
*BpBAM3b*	*Bpev01.c0443.g0028*	989	656–675	710–907	Cell membrane (0.9888)
*BpPXY*	*Bpev01.c0668.g0016*	1028	654–676	720–987	Cell membrane (0.9959)
*BpPXL1*	*Bpev01.c1696.g0001*	1000	627–649	699–969	Cell membrane (0.9977)
*BpCLV1*	*Bpev01.c2361.g0013*	986	647–666	704–971	Cell membrane (0.9936)

**Table 4 plants-11-01727-t004:** The features of the *B. pendula* WUSCHEL-related homeobox (WOX) family members.

Gene Name	Gene ID	Number of Amino Acid Residues in Peptide	Location of Homeobox Domain	Subcellular Localization (Likelihood)
*BpWUS*	*Bpev01.c0022.g0024*	274	25–85	Nucleus (0.9974)
*BpWOX13*	*Bpev01.c0022.g0148*	299	121–179	Nucleus (0.9933)
*BpWOX2*	*Bpev01.c0030.g0032*	177	19–52	Nucleus (0.9183)
*BpWOX1c*	*Bpev01.c0130.g0014*	240	31–66	Nucleus (0.9899)
*BpWOX1b*	*Bpev01.c0130.g0015*	128	60–96	Nucleus (0.8748)
*BpWOX11/12*	*Bpev01.c0441.g0004*	261	28–80	Nucleus (0.9984)
*BpWOX3*	*Bpev01.c0447.g0003*	200	8–67	Nucleus (0.9965)
*BpWOX4*	*Bpev01.c0640.g0004*	219	83–142	Nucleus (0.9907)
*BpWOX5*	*Bpev01.c0817.g0002*	168	30–88	Nucleus (0.9987)
*BpWOX1a*	*Bpev01.c1525.g0003*	334	65–124	Nucleus (0.9971)
*BpWOX8/9*	*Bpev01.c2517.g0001*	349	46–106	Nucleus (0.999)

**Table 5 plants-11-01727-t005:** The list of primers for the RT-PCR reaction.

Gene Name	Gene ID	Forward Primer (5′→3′)	Reverse Primer (5′→3′)	Ta, °C
*Ef1a*	*Bpev01.c0437.g0013*	TCCTTGAGGCTCTTGACTTG	ATACCAGGCTTGATGACACC	54
*GAPDH*	*Bpev01.c1040.g0016*	AGAATACAAGCCAGAACTCAAC	CTCTACCACCTCTCCAATCC	54
*BpCLE41/44a*	*Bpev01.c0016.g0065*	TGCTCCTCTTGCTTGTTACTC	GCCGATTGTTGTGTTGAAGG	55
*BpCLE41/44b*	*Bpev01.c0443.g0012*	TGGCGGCGTTGGCAAGTCC	TGGCAATGGCGGTTTACCTCTCC	60
*BpPXY*	*Bpev01.c0668.g0016*	GTCCTATACCGCCGAGATATG	GCACGCTACCTTCCAACG	55
*BpWOX4*	*Bpev01.c0640.g0004*	ACACCACCGACACTTCTTC	CCTTATACAGCATCTCCAATATCC	55

## Data Availability

Not applicable.

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
