# Peer review of "Changes in the Activity of the CLE41/PXY/WOX Signaling Pathway in the Birch Cambial Zone under Different Xylogenesis Patterns"

_plants, 2022, doi:10.3390/plants11131727_

Round 1
Reviewer 1 Report
1. Line 25, “can be studied within on tree trunk” , should be “can be studied within one tree trunk”?
2. I can’t see the main results in the abstract, the abstract just like the background.
3. 2.3,2.4,2.5, how to name the different genes in Betula? According to the Phylogenetic analysis?
4. It is better to show the BP, NFt,and Ft trees, at least the trunk.
5. 5.2. Plant sampling. Fell the tree? Or only pick a block from the growing trunk?
6. Y-axis title in each figure.
7. “other plant species” should be “Arabidopsis thaliana and Populus trichocarpa”.
8. “Seven genes encoding putative receptor like kinases homologous to TDR/PXY of other plant species were identified in B. pendula genome”, only six in figure 5.
9. Ling 135, Fraction 2 (from the side of the wood) included a differentiating xylem at all periods of sampling and contained xylem cells at the stage of expansion and formation of a secondary cell wall (Figure 1). line 262 “ We assumed that cell proliferation was observed in fractions 1 and 2 on all sampling dates.” why? Its activity correlates with proliferation process of BpWOX4 is not clear or can’t be a marker?
Author Response
Dear Reviewer!
We have revised and resubmitted our manuscript.
We are grateful to you for the analysis of our work and comments.
All changes in the manuscript are highlighted in green color.
Below you can faint the responses to the comments.
- 1. Line 25, “can be studied within on tree trunk”, should be“can be studied within one tree trunk”?
Our response. We corrected it.
I can’t see the main results in the abstract, the abstract just like the background.
Our response. We changed the abstract.
- 2.3, 2.4, 2.5, how to name the different genes in Betula? According to the Phylogenetic analysis?
Our response. Yes, we added it in Materials and Methods.
- It is better to show the BP, NFt,and Ft trees, at least the trunk.
Our response. We added it to the text.
- 5.2. Plant sampling. Fell the tree? Or only pick a block from the growing trunk? 5.2.
Our response. For microscopic analysis blocks, including the phloem, cambial zone and last 2-3 annual increments of wood were cut out (5 * 5 * 3 mm, length x width x height). For molecular genetic analysis «windows» of 6 * 8 cm were cut out of the trunk and the bark was separated from the wood. Also we added the additional figure.
- Y-axis title in each figure.
Our response. We added it to the text.
- “other plant species” should be “Arabidopsis thaliana and Populus trichocarpa”
Our response. We added it to the text.
- “Seven genes encoding putative receptor like kinases homologous to TDR/PXY of other plant species were identified in B. pendula genome”, only six in figure 5.
Our response. We added it to the text.
- Ling 135, Fraction 2 (from the side of the wood) included a differentiating xylem at all periods of sampling and contained xylem cells at the stage of expansion and formation of a secondary cell wall (Figure 1). line 262 “ We assumed that cell proliferation was observed in fractions 1 and 2 on all sampling dates.” why? Its activity correlates with proliferation process of BpWOX4 is not clear or can’t be a marker?
Our response. Level of WOX4 expression correlate strongly with cell proliferation (Etchells et al., 2013; Hirakawa et al., 2010) and used as genetic marker of this process (Alonso-Serra et al., 2019).
Reviewer 2 Report
This study may be of interest to potential readers working on the anatomy and/or molecular biology of plants. The authors provide information that may explain the distinctive anatomical features of silver birch varieties from a molecular perspective. The manuscript is mostly well written (although there are quite a few language details to be corrected). I found the terminological density of the molecular side of this text rather high (at least for the non-especialist like myself). But I believe molecular botanists will find it worth reading.

Author Response
Dear Reviewer!
We have revised and resubmitted our manuscript.
We are grateful to you for the analysis of our work and comments.
All changes in the manuscript are highlighted in green color.
Thanks a lot for your recommended corrections.
We found them very useful and of course we agree with them.
We changed the text due to your recommendations.
